# Carotenoids: Role in Neurodegenerative Diseases Remediation

**DOI:** 10.3390/brainsci13030457

**Published:** 2023-03-08

**Authors:** Kumaraswamy Gandla, Ancha Kishore Babu, Aziz Unnisa, Indu Sharma, Laliteshwar Pratap Singh, Mahammad Akiful Haque, Neelam Laxman Dashputre, Shahajan Baig, Falak A. Siddiqui, Mayeen Uddin Khandaker, Abdullah Almujally, Nissren Tamam, Abdelmoneim Sulieman, Sharuk L. Khan, Talha Bin Emran

**Affiliations:** 1Department of Pharmaceutical Analysis, Chaitanya (Deemed to be University), Hanamakonda 506001, Telangana, India; 2School of Pharmacy, KPJ Healthcare University, Persiaran Seriemas, Nilai 71800, Negeri Sembilan, Malaysia; 3Department of Pharmaceutical Chemistry, College of Pharmacy, University of Ha’il, Ha’il 55476, Saudi Arabia; 4Department of Physics, Career Point University, Hamirpur 176041, Himachal Pradesh, India; 5Department of Pharmaceutical Chemistry, Narayan Institute of Pharmacy, Gopal Narayan Singh University, Jamuhar, Sasaram 821305, Bihar, India; 6Department of Pharmaceutical Analysis, School of Pharmacy, Anurag University, Hyderabad 500088, Telangana, India; 7Department of Pharmacology, METs, Institute of Pharmacy Bhujbal Knowledge City, Adgaon, Nashik 422003, Maharashtra, India; 8Clinical Research Associate, Clinnex, Ahmedabad 380054, Gujarat, India; 9Department of Pharmaceutical Chemistry, N.B.S. Institute of Pharmacy, Ausa 413520, Maharashtra, India; 10Centre for Applied Physics and Radiation Technologies, School of Engineering and Technology, Sunway University, Bandar Sunway 47500, Selangor, Malaysia; 11Department of Biomedical Physics, King Faisal Specialist Hospital and Research Center, Riyadh 11564, Saudi Arabia; 12Department of Physics, College of Science, Princess Nourah Bint Abdulrahman University, P.O. Box 84428, Riyadh 11671, Saudi Arabia; 13Radiology and Medical Imaging Department, College of Applied Medical Sciences, Prince Sattam Bin Abdulaziz University, P.O. Box 422, Alkharj 11942, Saudi Arabia; 14Department of Pharmacy, BGC Trust University Bangladesh, Chittagong 4381, Bangladesh; 15Department of Pharmacy, Faculty of Allied Health Sciences, Daffodil International University, Dhaka 1207, Bangladesh

**Keywords:** carotenoids, neurodegeneration, reactive oxygen species, oxidative stress, oxidative stress, neuroinflammation

## Abstract

Numerous factors can contribute to the development of neurodegenerative disorders (NDs), such as Alzheimer’s disease, Parkinson’s disease, amyotrophic lateral sclerosis, Huntington’s disease, and multiple sclerosis. Oxidative stress (OS), a fairly common ND symptom, can be caused by more reactive oxygen species being made. In addition, the pathological state of NDs, which includes a high number of protein aggregates, could make chronic inflammation worse by activating microglia. Carotenoids, often known as “CTs”, are pigments that exist naturally and play a vital role in the prevention of several brain illnesses. CTs are organic pigments with major significance in ND prevention. More than 600 CTs have been discovered in nature, and they may be found in a wide variety of creatures. Different forms of CTs are responsible for the red, yellow, and orange pigments seen in many animals and plants. Because of their unique structure, CTs exhibit a wide range of bioactive effects, such as anti-inflammatory and antioxidant effects. The preventive effects of CTs have led researchers to find a strong correlation between CT levels in the body and the avoidance and treatment of several ailments, including NDs. To further understand the connection between OS, neuroinflammation, and NDs, a literature review has been compiled. In addition, we have focused on the anti-inflammatory and antioxidant properties of CTs for the treatment and management of NDs.

## 1. Introduction

Neurodegenerative diseases (NDs), which are caused by the slow death of neurons, are caused by proteins that stick together. The major NDs, such as Alzheimer’s disease (AD), Parkinson’s disease (PD), Huntington’s disease (HD), amyotrophic lateral sclerosis (ALS), and multiple sclerosis (MS), each have their own set of illness-specific causative variables and pathological features [1]. Even though these NDs can be caused by a number of different things, both how they start and how they get worse are tied to the death of neurons [2]. During the course of NDs, oxidative stress (OS), which arises due to the increased formation of reactive oxygen species (ROS), is one of the typical features that can be seen [3,4]. ROS is capable of attacking and causing damage to a variety of macromolecules found in live cells, including proteins, DNA, and lipids. It was discovered that the levels of ROS were elevated in the neurons of people who had NDs [5,6,7,8]. This led to the malfunctioning of mitochondria and the production of redox metals that reacted with oxygen, which ultimately resulted in the death of neuronal cells [9,10,11,12,13,14].

Neurodegeneration is characterized by changes in cytokine signaling, immune cell multiplication and movement, modified phagocytosis, and reactive gliosis. In the last 10 years, more and more studies have shown how important the immune system is in the onset and progression of NDs [15]. This has been the case in recent years. An efficient endogenous defense that protects the brain’s central nervous system (CNS) against pathogens and injuries is called neuroinflammation [16,17]. To be more specific, neuroinflammation is the stimulation of the neuroendocrine cells microglia and astrocytes into proinflammatory states [18]. In most cases, it is a beneficial process that works toward the elimination of risks and the restoration of equilibrium. Glial cells, which include microglia and astrocytes, play both a pro- and anti-inflammatory role and are involved in a variety of functions under both physiological and disease conditions. Some of these functions include phagocytosis, the discharge of steroids, the reduction of free radicals, and cellular repair [19].

Unfortunately, there are no pharmaceutical treatments yet that can stop or even slow down the progression of these deadly diseases [20]. Because of this, scientists are now putting more effort into finding naturally occurring chemicals that protect against certain diseases. Natural substances that have anti-inflammatory properties could be suitable candidates for the development of effective treatments [21]. This is because neuroinflammation plays a significant role in the beginning and progression of NDs [22]. Marine organisms are a major source of naturally occurring chemicals. Many of these chemicals have different structures from naturally occurring compounds that come from land [23,24,25]. Algae, which are classified as marine creatures, are one of the most important commodities found in the ocean [26,27]. There is a correlation between the intake of algae and a decreased risk of developing chronic degenerative diseases, according to epidemiological studies that compare the diets of Japan and the West [28]. There are many potent biological constituents that can be found in algae, including antioxidant compounds, peptides, vitamins, and mineral deposits, rich in soluble fibers, polyunsaturated fatty acids, polysaccharides, sterols, carotenoids, tocopherols, terpenes, phycobilins, phycocolloids, and phycocyanins. Algae are valued as sources of these essential phytochemical constituents.

This study focuses on the primary inflammatory processes that are associated with dementia as well as the potential of marine algae and some chemicals derived from marine algae to combat neuroinflammation in the CNS. We chose the most recent and important findings about the possible benefits of marine algae’s anti-inflammatory effects on neuroprotection.

## 2. Types of Neurodegenerative Diseases

Neuroglia, being the brain’s primary homeostatic and defensive component, plays a crucial role in the development of most neurological illnesses. Neuroglial responses are triggered by every kind of brain damage and are crucial to the development and manifestation of the diseased process. For instance, astrocytes play a pivotal role in the development of cerebral infarction and stroke. Ischemic core, indicated by the region of cellular pan-necrosis, develops rapidly, whereas the ischemic penumbra, which surrounds the zone of infarction, forms considerably more slowly. Astrocytes have a major role in determining the extent and speed of infarction zone growth via the penumbra, which might take days. In most cases, astroglia can withstand ischemia and excitotoxicity without suffering damage. Likewise, oligodendroglia plays a crucial role in white matter ischemic injury. Ischemia to the white matter causes disruptions of nerve fibers and serious functional deficits because the death of oligodendrocytes leads eventually to axonal disintegration. Ischemia and glutamate excitotoxicity are particularly hazardous to oligodendrocytes and their progenitors, oligodendroglial cells. Oligodendroglial ion homeostasis and Ca^2+^ overload toxicity are lost after even brief durations of anoxia/ischemia. In chronic NDs, neuroglia also emerges as an important actor. Certain forms of epilepsy were shown to cause dramatic changes in astrocytic shape and function. Astroglial K^+^ buffering, water transport, and glutamate accumulation were all found to be impaired in animal models of epilepsy [29]. Depression and other mental diseases have both been linked to apoptosis and chronic astroglial atrophy [30]. As the CNS’s innate immunological and phagocytic cellular components, microglial cells are always implicated in some way in every neuropathology [31]. The response of microglial cells to brain lesion and the development of neuroinflammation are both determined by a complicated and multistage process of microglial activation that is set off by brain injuries of varying origin.

## 3. Role of Carotenoids and Background of Neurodegenerative Diseases

CTs, in the broadest sense, are organic substances essential to human growth and health. Vitamins are essential nutrients that the body is unable to produce on its own, at least not in adequate amounts. They are essential nutrients that can only be consumed in food. CTs play important roles, often as antioxidants or enzymatic cofactors. Water-soluble CTs cannot be absorbed into the body and must be consumed on a daily basis as an exogenous source. On the other hand, the body uses fat-soluble CTs right away and does not store them until they are needed. An active material that can persist on its own and has one or even more unpaired electrons is considered a free radical. Free radicals with an oxygen atom (O) or its equivalent that react more strongly with other molecules than with O_2_ are known as ROS and reactive nitrogen species (RNS) [32,33]. Free radicals derived from oxygen, including superoxide anion and the hydroxyl radical, are among the most significant contributors to a wide variety of disease conditions. The generation of radicals in the body can take place via a variety of different pathways that involve both endogenous and exogenous components. The attachment of a free electron to oxygen results in the production of superoxide anion, and there are multiple pathways that can result in the production of superoxide in living organisms. CTs provide, in most cases, a considerable benefit for the treatment of NDs. Both CTs that are water- and lipid-soluble can play a significant role in warding off PD and AD. CT supplementation has been shown to be effective in preventing α-synuclein toxicity. In addition, CTs are known to safeguard the dopamine transporter through the protective effects that they have. An increased intake of CTs can also help protect against the amyloid and tau diseases that might develop over time. CTs have also been shown to have a curative effect on disorders such as MS, HD, and prion disease [34,35].

In terms of quantity, CTs are the most important lipophilic antioxidant found in the brain. It is generally accepted that the primary role of CT is to defend lipids against the damaging effects of OS. A number of studies have attempted to test the hypothesis that increased OS leads to increased consumption of CT in the brains of AD patients. This should lead to lower levels of the CTs in the brain and/or cerebrospinal fluid (CSF). The results of these studies have been mixed. There was no link discovered between the use of carotenoid in either dietary or supplementary form (or in either or both forms) and the risk of AD. In this particular research project, unfavorable findings about the use of CTs and vitamin C (Vit C) were discovered. CT and Vit C supplementation, on the other hand, were discovered to have a considerable protective impact against vascular dementia. In addition, the usage of either CT or Vit C supplements alone at baseline was related to superior cognitive function at follow-up in participants who were free of dementia at baseline [36].

## 4. Effects of Reactive Oxygen Species (ROS) and Reactive Nitrogen Species (RNS) into Oxidative Stress (OS)

In the late 1960s, McCord and Fridovich demonstrated that superoxide free radical anion (O^2•−^) could be produced enzymatically in mammalian tissues. They also showed that superoxide dismutase (SOD) catalyzed the dismutation of O^2•−^. These findings were significant because they showed that mammalian tissues are capable of producing O^2•−^ [37]. When oxygen is present, certain molecules, such as flavine nucleotides and thiol compounds, undergo oxidation, producing superoxide. The presence of transition metals, such as iron or copper, dramatically accelerates these reactions. The conversion of oxygen into water is carried out via the electron transport chain, which is located in the inner mitochondrial membrane. During the course of the process, free radical mediators are produced; these radicals are typically strongly attached to the various components of the transport chain. However, the mitochondrial matrix is subject to a steady loss of a few electrons, which ultimately culminates in the production of superoxide. It is also possible that the endothelial surface is constantly producing superoxide anion in order to neutralize nitric oxide; that other cells are producing superoxide in order to control cell proliferation and differentiation; and that phagocytic cells are producing superoxide during the oxidative burst. In every biological system that produces the superoxide anion, hydrogen peroxide will also be produced as a byproduct of a spontaneous dismutation process. In addition, several enzymatic processes have the potential to directly create hydrogen peroxide. Quite frequently, the involvement of ROS in these activities is established in an indirect fashion through the utilization of antioxidant molecules. In fact, the half-life of ROS in the form of O^2•−^ is rather short, and determining how much of it is produced is a difficult process. This is especially true in nonphagocytic cells such as the endothelium. On the other hand, ROS that originate from the endothelium appear to play a significant role in the pathophysiology of a variety of processes, such as aging and atherosclerosis. Barbacanne et al. concentrated their efforts on the creation of extracellular O^2•−^ due to the fact that the specificity of the signal is supplied by the use of SOD and that this control cannot be reached within the cells themselves [38].

Immune cells such as macrophages and neutrophils release nitrogen monoxide and superoxide into phagocytic vacuoles at the same time. This is how peroxynitrite is made, which kills bacteria that have been taken into the cell. Since it was found that mammalian cells are capable of producing the free radical nitric oxide (NO), there has been an astonishing surge in the amount of study that is being performed in all areas of biology and medicine related to these subjects. Since its primary identification as an endothelial-derived relaxing factor, NO has evolved as a key signaling device that regulates practically every important cellular function. Additionally, NO has emerged as a powerful mediator of cellular damage in a broad variety of circumstances. New data suggest that the majority of the cytotoxicity that has been attributed to NO is really attributable to peroxynitrite. Peroxynitrite is formed as a result of the diffusion-controlled interaction that takes place between NO and another free radical known as the superoxide anion. Elevated concentrations of NO and neurotrophin-3 (3-NT) have been documented in a range of human skin illnesses, including skin malignancies, systemic lupus erythematosus, psoriasis, urticaria, and atopic dermatitis. Other skin diseases include atopic dermatitis and urticaria. The many endogenous sources of ROS and RNS are broken down and presented in Figure 1.

## 5. Game Changer Carotenoids for Parkinson’s Disease

Degradation of neurotransmitter receptors in the midbrain leads to the motor and non-motor symptoms of PD. Following in the footsteps of AD, PD is thought to be the most prevalent form of progressive neurodegenerative illness. The loss of dopaminergic neurons in the substantia nigra pars compacta (SNpc) and a decreased level of dopamine (DA) in the striatum are the two primary symptoms of PD. Rai et al. investigated the effect of several doses of ursolic acid (UA) on neurocognitive measures, and they also looked at the enzymatic antioxidant systems. Epileptogenesis and neurodegeneration are facilitated by neuroinflammation and oxidative stress, which in turn promotes persistent epilepsy and cognitive deficiency. The neuroprotective effects of ursolic acid (UA) have been demonstrated in previous research. UA inhibits inflammation and oxidative stress. Both the amount of dopamine and the expression of tyrosine hydroxylase (TH) in the substantia nigra (SN) were measured using immunohistochemical method. The level of dopamine was compared to its metabolites [39]. Catecholamines are a class of hormones that are produced in a multi-step process that tyrosine hydroxylase is a component of. Dopamine is a catecholamine that may be produced from the amino acid tyrosine. ROS and RNS, the byproducts of which were detected in the substantia nigra and striatum of human PD post-mortem brains, are known to cause damage to biomolecules such as lipids, proteins, and DNA [40]. In NDs, the oxidation of lipids and proteins can therefore cause a loss of membrane integrity, which can lead to the inactivation of enzymes, which ultimately results in cell death [41].

OS is a significant contributor to the course of PD, and vitamin A (Vit A), along with its derivatives such as retinoic acid (RA), has been shown to have powerful antioxidative action [42]. Animal and plant products can both be a source of Vit A. There are three different forms of Vit A: retinol, retinal, and retinoic acid (RA). Of these, RA demonstrates the highest level of biological activity. There are two important forms of RA that may be found in nature: 9-cis-RA and all-trans-RA (ATRA) (Figure 2) [43]. The preservation of one’s vision, one’s capacity to grow, and the integrity of one’s epithelial and mucous tissues are among the fundamental biological roles of Vit A [44].

Quite pertinently, CTs has been shown about the remediation to various childhood sickness (Table 1) with diseases that spread through the lungs, such as pneumonia and measles in children, or with diseases that spread through the digestive tract, such as imbecilic gastroenteritis and palm, big toe, and throat illness in children. Scientists have also indicated a relationship between CTs and PD; however, despite all of the efforts that have been made, it is still not apparent if a shortage in CTs are the cause of PD or a result of PD [45]. CTs are known to exert their therapeutic benefits through a variety of methods, including as the regulation of cell development, the modification of gene expression, and the enhancement of immunological response. Autoantibodies, or immunoglobulin A (IgA) directed against one’s own proteins, are often found in the plasma of healthy people. Some of these autoantibodies play pathogenic roles in systemic or tissue-specific autoimmune disorders such rheumatoid arthritis and lupus. On the other hand, it is believed that their principal protective mechanism is owing to the high antioxidant qualities that they possess. These features have the capability to competently calture free radicals and minimize the danger of OS. With over 600 different CTs have been discovered in nature, and the pattern of the vast majority of them is that of a polyisoprenoid. This is achieved by the linking of the terminal tails of two different C20 diphosphate molecules. The electrons are dispersed across the entirety of the system as a result of the presence of a lengthy polyene chain that contains anywhere from 3 to 15 conjugated double bonds (Figure 3). Because of the lipophilicity of CTs, their digestion follows a process that is analogous to that of fats consumed in the diet. This process is kicked off by the release of CTs from the matrix of the meal, which occurs as a result of chewing and the activity of digestive enzymes. After the CTs have been released, they are subsequently integrated into lipid droplets inside gastric emulsions. After this, they are solubilized in mixed micelles constituted of dietary lipids, bile salts, and biliary phospholipids by the activity of lipolytic enzymes in pancreatic juice [46].

During usual conditions, inflammation serves as a protective mechanism for tissues, defending them against both endogenous and exogenous harm. There are a number of different agents and circumstances that are known to be capable of causing inflammation. These include microbial and viral diseases, immunological illnesses, sensitivity to allergen or hazardous materials, or even metabolic derangements, which can include overweight. A pathological disease known as chronic inflammation, chronic inflammation is defined by a continuous active inflammatory response and tissue damage. In the pathophysiology of chronic inflammation, several different immune cells, such as macrophages, neutrophils, and eosinophils, are either directly engaged or indirectly implicated through the generation of inflammatory cytokines. According to the research that has been conducted, there is a widespread belief that chronic inflammation might be a primary factor in the development of cancer as well as the expression of the aging process. In addition, the results of a number of studies point to the possibility that chronic inflammation plays a significant part in a wide variety of age-related disorders, such as diabetes, cardiovascular disease, and autoimmune disease. OS is induced during the inflammatory process, which also decreases the cellular capacity for antioxidant defense. The overproduction of free radicals causes a reaction with the fatty acids and proteins in the cell membrane, which irreversibly impairs the function of the membrane. Free radicals may also cause mutations and damage to DNA, which can be a role in the development of age-related diseases as well as cancer. This makes free radicals a potential risk factor for both of these conditions. Acute inflammation is separated from chronic inflammation by two distinct phases. In most cases, acute inflammation is helpful to the host because it assists in the process of reestablishing normal homeostasis. For instance, it does this by digesting invading germs. This stage of inflammation does not last long. The likelihood of developing cancer increases if inflammation continues for an extended period of time. There are two distinct types of inflammation: chronic and acute inflammation. Inflammation results in the recruitment of mast cells and leukocytes to the site of injury, which causes a “respiratory burst” owing to an increased intake of oxygen and, as a result, enhanced generation and buildup of ROS at the damage site. Proinflammatory cytokines, in contrast side, are responsible for the production of soluble factors. These soluble mediators include metabolites of arachidonic acid, cytokines, and chemokines, and their actions include further recruiting inflammatory cells to the site of destruction and generating extra superoxide radicals. Dissolved intermediaries produced by inflammatory cells include cytokines, chemokines, and arachidonic acid metabolites (such as prostaglandins), all of which play a role in the recruitment of macrophages and serve as important key activators of various signaling pathways streams and transcription factors such as NF-κB and nuclear factor E2-related factor 2 nuclear factor E2-related factor 2 (Nrf2) (Figure 4).

The Keap1-Nrf2 pathway is an essential component of the cell’s defense mechanism against ROS, which can be generated either endogenously or exogenously. It was advocated by Trippier et al. that novel medicines be developed to address the underlying pathophysiologies of NDs [47]. When addressing the possible therapeutic benefits of antioxidants, it is important to take into account the role that the Nrf2 plays in signaling gene expression in response to cellular stress. The transcription factor Nrf2 is responsible for the activation of a number of genes, the products of which are essential in the process of lowering the amount of OS experienced by the cell. Changing the cysteine residues of Keap1 causes a disruption in the binding of Nrf2, which in turn enables Nrf2 to translocate into the nucleus. After entering the nucleus, Nrf2 forms heterodimers with members of the family of transcription factors known as small Maf (sMaf), binds to the ARE and eventually causes gene expression to occur (Figure 5).

## 6. Carotenoids as Anti-Apoptotic Agent

Illnesses of the brain that lead to the death of large numbers of neurons (apoptosis) also have a significant influence on the development of NDs. Apoptosis can be triggered in cerebral cells in response to neurotoxic activation in one of three ways: death receptor facilitated (extraneous), mitochondria-mediated (inherent), or an amalgamation of both. OS, excitotoxicity, and neuroinflammation are examples of neurotoxic stimulation. Activated death receptors such as tumor necrosis factor receptor 1 form a trimer, which then recruits adapter proteins. These adapter proteins include cellular inhibitors of apoptosis (cIAP) 1 and 2, TNFR1 accompanying death domain protein (TRADD), and Fas-associated death protein (FADD). To prevent mitochondrial dysfunction and cell death, tBid is transported to the mitochondrial membrane, where it interacts with and captures multidomain proapoptotic Bcl-2 proteins, including Bcl-2 connected X protein (Bax). Oligomerization of Bax with itself or other proapoptotic proteins, such as Bak, opens the mitochondrial membrane and triggers the apoptotic cascade (Figure 6). Cytochrome c, an indispensable constituent of the electron carriage chain, is localized on the internal mitochondrial membrane in healthy cells. However, because of its hydrophilicity, cytochrome c is released into the cytoplasm after mitochondrial outer membrane permeabilization. Apoptotic protease activating factor (Apaf) 1 and dATP bind to cytochrome c in the cytoplasm to generate an apoptosome. It is suspected that mitochondrial pathways are implicated in the etiology of PD. When heroin addicts got PD after being exposed to 1-methyl-4-phenyl-1,2,3,4-tetrahydropyridine, which is broken down by monoamine oxidase into a complex I inhibitor and free radical generator, mitochondria were thought to be the cause of PD. Martin et al. did trials to see if their theory that the mitochondrial permeability transition pore (mPTP) is involved in the process of getting sick was true. They did this by studying a model of human Syn in transgenic mice. [61]. Neurotransmitter receptors in the brains of young mice experience apoptotic degradation. A mutant form of Syn was shown to be related through dysmorphic neuronal mitochondria and the binding of voltage-dependent anion networks.

The process of apoptosis is also very crucial in AD. The deposition of amyloid causes apoptosis to be triggered not only by death receptors mediating extrinsic apoptosis but also by mitochondria mediating intrinsic apoptosis. The amyloid precursor protein (APP) is cleaved to produce the A peptide, which is then metabolized by secretases (α, β, and γ secretase) to produce non-amyloidal and/or amyloidal byproducts. The triggering of apoptotic signaling is what causes the morphological and physiological degradation of the brain that occurs during the process of neurodegeneration. To this day, there is no therapy that can reverse the pathology of AD. In this review, we discuss the functions that several CT pigments play in inhibiting the process of neuronal death.

The pharmaceutical and food sectors are always looking for new and interesting sources of bioactive chemicals, and microalgae are among the most promising of these sources. CTs stand out among these bioactives because of the positive impact they have on health, such as their anti-inflammatory, neuroprotective, or antioxidant capabilities. Microalgae extracts were shown to have a medium and specific cholinergic inhibitory activity, as well as significant antioxidant and anti-inflammatory capabilities, with particular emphasis on the extracts of *N. oceanica* and *T. lutea*. These findings were published by Gallego et al. [62].

The presence of neuroinflammation is one of the defining characteristics of NDD. Microglia, neutrophils, astrocytes, and macrophages are only a few of the cell types that play an important part in the process of neuroinflammation. Table 2 contains a discussion on the role of CTs as neuroprotective agents as well as the structure and targeted pathway of NDs. Marine species produce an abundance of bioactive proteins, which are employed extensively in research due to the fact that they have the ability to treat NDs and the activities of drugs they have. As people live longer, their likelihood of developing NDs also rises. The development of more advanced technology has aided in the diagnosis of conditions that are associated with the central nervous system. Modern research aims to develop potential compounds derived from marine sources for use in the treatment of neural disorders, with a focus on the CNS and stimulating neurogenesis.

## 7. Clinical Studies

CTs are a kind of pigment that is soluble in fat and may be found in fruits and vegetables. They are responsible for the vibrant colors of these foods, including yellow, orange, red, and green. There is some evidence that higher plasma CT levels improve immune response and lower the risk of contracting infectious diseases. Serum CT levels are one of the greatest predictors of whether or not an individual consumes fruit and vegetables. It is widely believed that antioxidant vitamins have a beneficial effect on pulmonary function and that reduced pulmonary function is one of the most important factors in determining the risk of death in the general population. Recent research has placed a significant amount of emphasis on determining the function that the equilibrium between the body’s oxidative load and its antioxidant potential plays in the etiology of airway blockage [63]. There has not been a great deal of study performed on the connection between plasma CT and respiratory symptoms in all aged populations and have a more compromised state. In two major preventative studies conducted on high-risk populations, it was discovered that those participants who received big amounts of beta-carotene supplementation had an increased chance of developing lung cancer.

It has been known for a long time that NDs might be caused by the harmful effects of OS. AD and PD have been linked to OS, which includes free radicals [71]. Because of its physiological role, OS may contribute to the gradual deterioration and late development of NDs. The human brain’s CNS is particularly susceptible to damage from free radicals. This highly oxygenated organ relies on the oxidative metabolism of the mitochondrial respiratory activity for its primary energy supply. Because of insufficient levels of catalase, superoxide dismutase (SOD), and glutathione peroxidase (GPx), the superoxide anion (O^2•−^) and hydrogen peroxide (H_2_O_2_) generated by live tissue under OS are not quickly neutralized. In addition, polyunsaturated fatty acids, which are abundant in brain membrane lipids, are particularly vulnerable to lipid peroxidation caused by free radicals. Vitamins, minerals, and phytochemicals are just a few examples of the natural food-derived components that have garnered much attention as of late due to their reputation as healthy, useful snacks. Antioxidant Vit C may be stored in water. Several studies have shown that lowering the risk of developing AD might be as simple as increasing one’s vitamin C consumption. A significant number of studies have discovered favorable correlations between the use of Vit C or foods rich in Vit C content and improved pulmonary function [64]. There was an elevated risk of lung disease in the group that received high doses of beta-carotene supplements, according to the findings of two major preventive studies conducted on massive populations. To this day, more than 600 different types of CTs have been identified, and several of these CTs (such as α-cryptoxanthin, lutein/zeaxanthin, α-carotene, and lycopene) are known to possess potent antioxidant properties [65]. Even though the consumption of food products rich in natural antioxidants is involved with other health habits that may impact lung function, the results of studies that focused on dietary intake are inconclusive with regard to the true nature of the interaction between these antioxidants and lung function [66]. First place goes to zeaxanthin, next to lutein, and finally α-cryptoxanthin when it comes to the importance of a CTs role in maintaining healthy lung function. When each component was analyzed on its own, beta-carotene, β-cryptoxanthin, lutein/zeaxanthin, and retinol showed a positive inter-relationship with forced expiratory volume (FEV). Investigations that focused on dietary intake raise issues regarding the real nature of the connection between these CTs and pulmonary performance. This is due to the fact that the intake of foods high in carotene is connected to other lifestyle choices that may impact lung function [67]. There has not been a parallel examination of the interplay of serum CT and its connection with lung capacity in epidemiologic studies.

Peng et al. wanted to examine the effects of 1 M β-carotene on antioxidant status in ethanol-treated rat hepatocytes and study probable anti-apoptotic mechanisms of beta-carotene in preventing ethanol-induced cytotoxicity. They constructed their experiment to do both of these things [72]. According to the findings, treatment with ethanol reduces the number of viable cells found in rat hepatocytes through the induction of oxidative stress. The expression of caspase-9 and caspase-3 was inhibited by 1 M β-carotene, which resulted in a reduction in OS and prevention of cell death caused by ethanol. There is evidence that hydrophobic bile acids have a role in the etiology of cholestatic liver illnesses via pathways that involve OS and mitochondrial dysfunction. In rat hepatocyte suspensions, the cytotoxicity caused by bile acid is reduced when antioxidants are present. Gumpricht et al. investigate the possible protective function of β-carotene, a recognized lipophilic antioxidant that is decreased in patients with cholestasis, besides bile acid-induced hepatotoxicity. β-carotene is reduced in individuals with cholestasis. A low amount of β-carotene (50 M) formed, such as antioxidant and anti-apoptotic defense, but with low inhibition against cell death. This finding suggests that a high amount of β-carotene may have discussed supplementary cytoprotection that was not straight connected to its antioxidant behavior [73].

Studies are now being conducted to investigate whether or not CTs have the ability to influence the formation of tumors. Clinical experiments have shown that supplementing chronic smokers or asbestos workers with β-carotene raises the risk for lung cancer, despite data from epidemiological research showing that high consumption of vegetables is rich. Interest in understanding how CT compounds work in living organisms has been rekindled in light of these seemingly contradicting results. In this review, we present evidence that CTs can have mitogenic and apoptotic effects, and we offer support for the idea that these chemicals may operate as anticarcinogens or procarcinogens via a redox mechanism. Since Palozza et al. published the in vitro and in vivo antioxidant and pro-oxidant actions of CTs, we have been paying close attention to the correlation between cell proliferation and redox state [74].

## 8. Safety and Toxicity of Carotenoids

The expansion of our understanding of the biology of CTs has made way for new research avenues that branch off in a variety of directions. Due to the fact that CTs are such effective scavengers of singlet oxygen in addition to being competent scavengers of other RNS, much research has been performed on the possible function that CTs may play as antioxidants. Primate species and ruminant animals are the only types of animals known to store considerable quantities of CTs in their bodies, with a few isolated outliers. Only humans and cows, among all animals, acquire yellow fat as they become older. Most other animals will not. Nutritional CT, for instance, is perfectly safe to eat in quantities that are far larger than those that are seen in persons who consume a typical diet [75,76]. As a direct consequence of this, the European Food Safety Authority (EFSA) has provided guidelines for the acceptable daily intake of certain CT pigments, which are also referred to in certain contexts as recommended daily intakes (ADI). For instance, the acceptable daily intake (ADI) for lutein is determined to be 1 mg/kg/day when it is produced from the floral essence of *Tagetes erecta* and used as an additive, but the ADI for lycopene is 0.5 mg/kg/day [77]. Zeaxanthin is a nutritious CT with a significant quantity of safety data gathered from regulatory studies that form the foundation of its safety review. These studies were conducted to ensure that the CTs was safe [78]. There is no indication of prenatal toxicity or teratogenicity in rats or rabbits that were subjected to developmental toxicity trials at doses of up to 1000 or 400 mg/kg bw/day, respectively. These dosages were based on the animals’ body weights. In a battery of in vitro and in vivo testing for mutagenicity, compounded zeaxanthin did not show any signs of being mutagenic or clastogenic. Several products of CT degradation are put to use as scents in the perfume business. These include damascones and damacenones, among others. According to the findings of the study, taking a combination of beta-carotene and retinyl palmitate on a daily basis may have a detrimental impact on the prevalence of lung cancer and may also raise the risk of dying from vascular complications, lung diseases, and other illnesses that can be brought on by a lifetime of smoking or exposure to asbestos [79].

## 9. Concluding Remarks

In this piece, we take a look at the most recent information about the positive impact that CTs have on disorders that affect the nervous system. In the context of NDs, a cumulative amount of research has established that nutritional CTs defend neurons through multiple mechanisms. These mechanisms include the quelling of ROS, the up-regulation of antioxidant enzyme coordination, and the anti-neuroinflammatory effects. Furthermore, there has been a steady rise in the number of research works available in current years that investigate the possible connection between CTs and NDs. It is remarkable that animal and cell culture prototypical lessons have only just begun to be dynamically led, and the results of these model studies provide strong support for the assumption that consumption of CTs may have therapeutic significance in either preventing or ameliorating a variety of NDs. Recent studies that were carried out using animal models have shed light on the neuroprotective benefits that dietary CTs, including astaxanthin, crocin, crocetin, and fucoxanthin, have on NDs. Our comprehension of the connection between the acceptance of CTs and NDs has grown as a direct consequence of this discovery.

The pathogenesis of NDs as well as the therapeutic response to NDs, is still not completely understood, despite the progress that has been achieved in research on NDs. CTs have the potential to be developed into drugs for the treatment of NDs as well as for the prevention of such conditions. Although the results obtained in vitro and in vivo are promising, more clinical research on humans that is carefully carried out is necessary before drawing any conclusions on the full potential of NDs. CTs perform the role of antioxidants, inhibiting the upstream causes of apoptosis and ROS-associated mitochondrial dysfunction. CTs prevent the loss of energy and, as a result, the activation of apoptotic chemicals that are produced from mitochondria. Determining the preventive and therapeutic activities of CTs in neurodegeneration will require greater research into distinguishing the exact molecular pathways of neuronal death that are involved, as well as ongoing translational investigations in human subjects.

## Figures and Tables

**Figure 1 brainsci-13-00457-f001:**
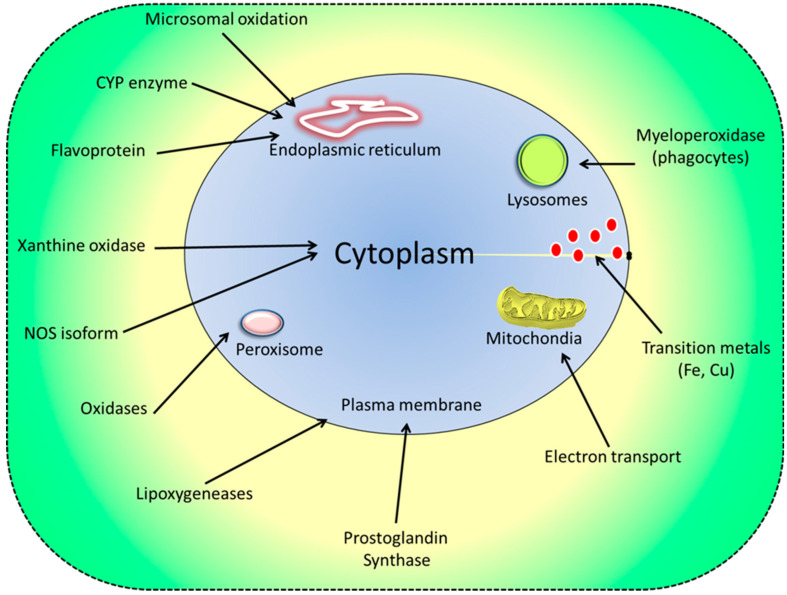
Endogeneous sources of ROS and RNS. Where, ROS, reactive oxygen species; RNS, reactive nitrogen species; CYP, cytochrome; NOS, nitric oxide synthase.

**Figure 2 brainsci-13-00457-f002:**
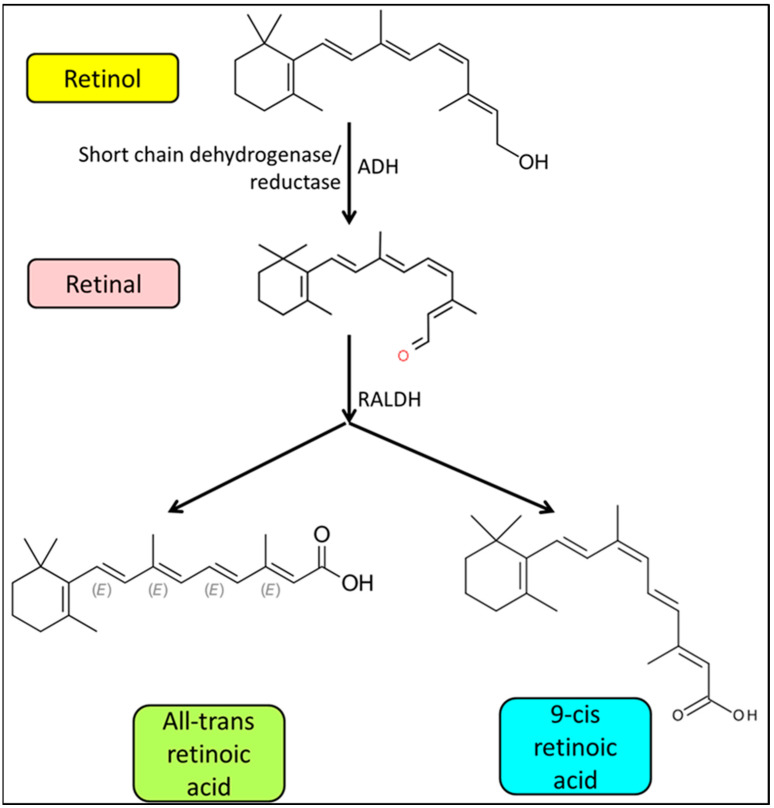
The oxidative process that is required to convert inactive retinol into useful retinoic acid takes place in two stages. Where, ADH, alcohol dehydrogenases; RALDH, retinaldehyde dehydrogenase.

**Figure 3 brainsci-13-00457-f003:**
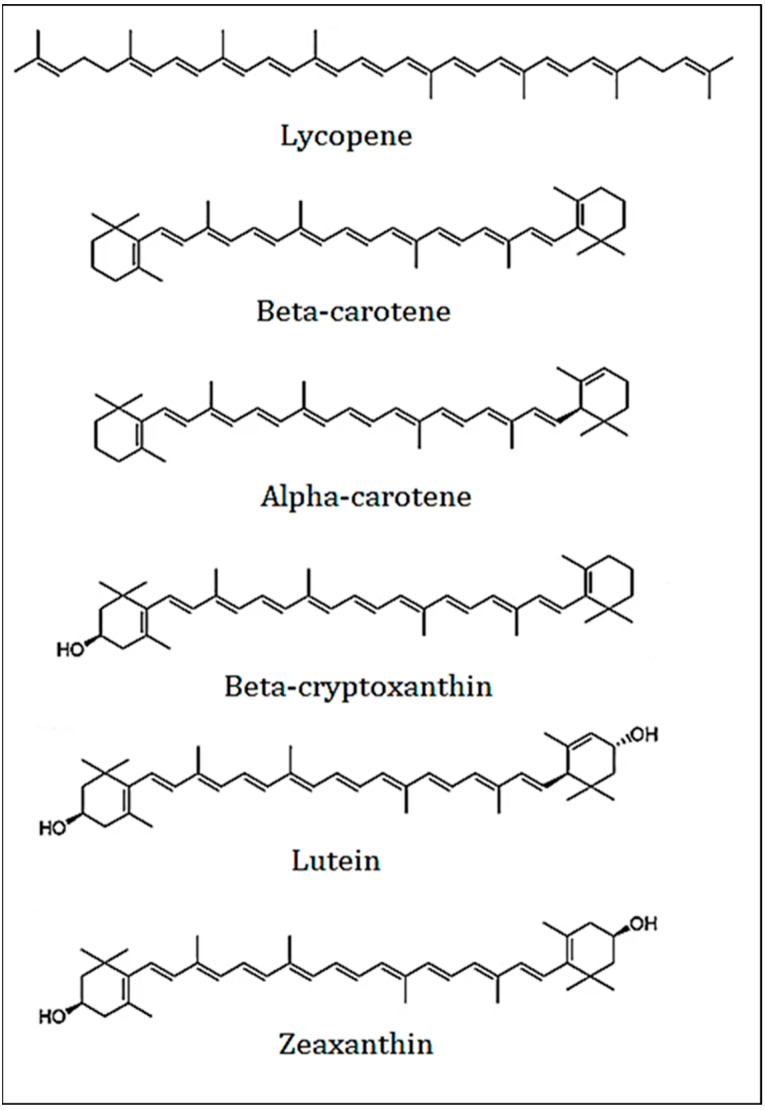
The constituent molecules that make up the chemical structure of various common plasma carotenoids.

**Figure 4 brainsci-13-00457-f004:**
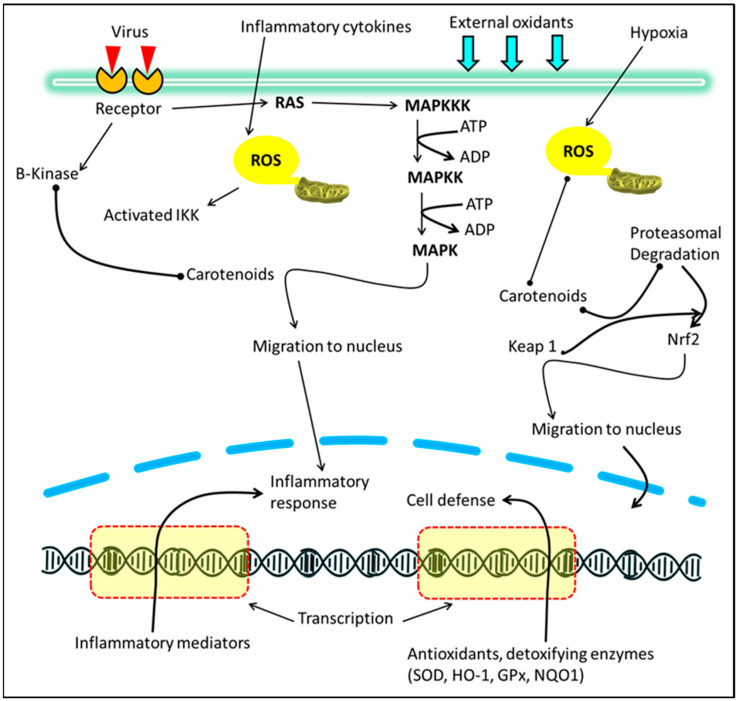
Carotenoids and the signaling pathways involved in inflammation. MAPK, mitogen-activated protein kinase; ROS, reactive oxygen species; Nrf2, nuclear factor erythroid 2–related factor 2; SOD, superoxide dismutase, RAS, rat sarcoma.

**Figure 5 brainsci-13-00457-f005:**
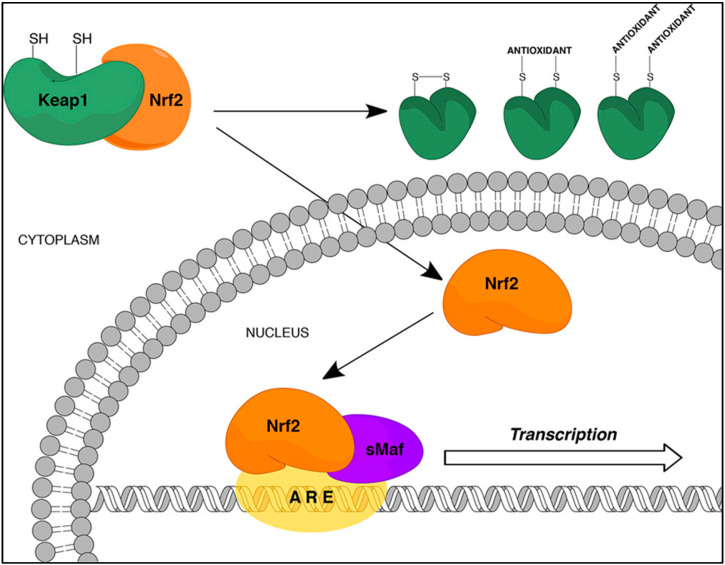
The mechanism by which Nrf2 promotes expression of genes that contain the ARE element. The migration of Nrf2 to the nucleus is made possible by the disruption of the Keap1–Nrf2 complex [47].

**Figure 6 brainsci-13-00457-f006:**
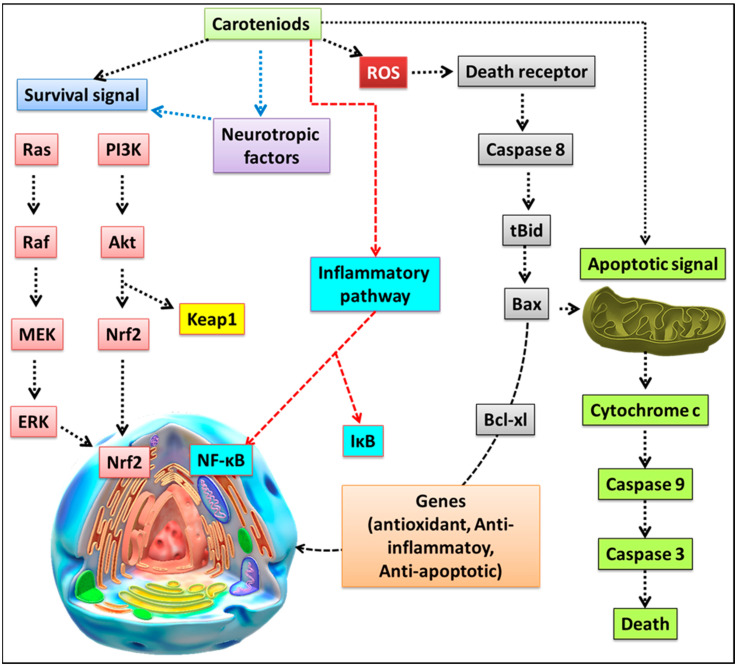
Carotenoids’ protective functions during neuronal death. Reactive oxygen species (ROS) can initiate apoptosis via the death receptor pathway and the mitochondrial route. (Reactive oxygen species (ROS); mitogen-activated and extracellular-signal-regulated kinase kinases (MEK); extracellular-signal-regulated kinases (ERK); nuclear factor (erythroid-derived 2)-such as 2 (Nrf2); impaired antioxidant (Keap1); small Maf proteins (sMaf); antioxidant response element (ARE); phosphoinositide 3-kinase (PI3K); protein kinase B (Akt); nuclear factor-κB (NF-κB); membrane-targeted death ligand (tBid); apoptosis regulator (BAX); B-cell lymphoma (Bcl-xl)).

**Table 1 brainsci-13-00457-t001:** Treatment of many contagious childhood illnesses with carotenoids.

Role of Carotenoids	Disease	Analysis Method	Model Taken	Ref.
Mortality reduction	Measles	Meta-analysis	Human	[47]
Morbidity and mortality reduction	Measles	Systematic review and meta-analysis	Human	[48]
Mortality reduction	Measles	Meta-analysis	Human	[49]
Mortality reduction	Measles	Randomized double-blind controlled trial	Human	[50]
Increasing antibodies	Pneumonia	Randomized controlled trial	Mice	[51]
Relieving medical symptoms	Pneumonia	Meta-analysis	Human	[52]
Morbidity and mortality reduction	Diarrhea	Systematic review and meta-analysis	Human	[48]
Increase intestinal immunoglobulin A (IgA) production and mucosal immune function	Diarrhea	Randomized controlled trial	Mice	[53]
Reduce morbidity	Diarrhea	Randomized double-blind controlled trial	Human	[50]
Morbidity and mortality reduction	Enteric infection	Randomized controlled trial	Mice	[54]
Reduce morbidity	Malaria	Randomized double-blind controlled trial	Human	[55]
Reduce morbidity	Malaria	Randomized controlled trial	Human	[56]
Reduce morbidity	Malaria	Randomized double-blind controlled trial	Human	[57]
Drive up Ig production and beef up your immune system’s ability to fight off viruses	Hand, foot, and mouth disease	Cross-sectional observation and study	Human	[58]
The increase in type 1 interferon and its ability to suppress viral replication	Mumps	In vitro controlled experiment	Live cells	[59,60]

**Table 2 brainsci-13-00457-t002:** Carotenoids, their representative high-yield sources, and their structures, coupled with the NDs pathways that they target.

Carotenoids	Source	Concentration	Pathways for Neuroprotection	Ref.
α-Carotene	*Daucus carota Cucurbita* spp.	58.8% of total carotenoid	Decrease AD and PD risk in people	[2]
β-carotene	*Dunaliella salina* *Elaeis guineens* *Saccharomyces cerevisiae* *Chlorella saccharophila* *Odontella aurita* *Chlorella zofingiensis(CZ-bkt1)*	14% of dry weight 40 mg/g 5.9 mg/g dry weight 4.98 mg/g 18.47 mg/g 34.64 mg/L	Neuropathic inhibition PD and ALS enzyme activation	[62]
Lycopene	*Solanum lycopersicum* *Sachharomyces cerevisiae* *Haematococcus pluvialis* SAG 19-a	100 µg/g 54.63 mg/g DCW 1.4 mg/g DCW	ARE activation and phase II enzyme induction PD anti-inflammatory drugs target autophagy	[63]
Crocin	*Crocus sativus* *Gardenia jasminoides Ellis*	0.43 g/L 8.4 mg/g dry powder	Microglial stimulation and excitotoxic pathway inhibition AD and PD antioxidants and antineuroinflammatory pathway NMDA inhibitor	[64]
Trans-crocin 4	*Stigma of Crocus sativus*	102 mg/g	Inhibition of Aβ-aggregation by inhibition of JNK/p38 pathway	[65]
Bixin	*Bixa orellana*	22 mg/g dried seeds 5–6% weight of seeds	Attenuation of inflammatory pathway by inhibiting NF-κB activation	[66]
Lutein	*Marigold* *Chlorella vulgaris* *Chlorella zofingiensis(CZ-bkt1)*	3.823 µg/g 3.36 mg/g 33.97 mg/g	Targets autophagy and apoptosis pathway Inhibition of NF-κB pathway	[67]
Zeaxanthin	*Chlorella zofingiensis(CZ-bkt1)* *Chlorella saccharophila* *Dunaliella tertiolecta*	36.79 mg/g 11.2 mg/g 8 mg/L	Activate antioxidant and anti-inflammatory	[68]
β-Cryptoxanthin	*Cucurbita moschata Duch.*	3.4% of total carotenoids	Inhibition of oxidative damage	[68]
Fucoxanthin	*Nitzschia laevis* *Phaeodactylum tricornutum*	2.24 mg/g dry weight 15.7 mg/g dry weight	Targets P13k/Akt pathway Activation of autophagy pathway	[69]
Fucoxanthinol	*Nitzschia laevis*	4.64 mg/g dry weight	Inhibition of P13k/Akt and MAPK pathway	[70]

## Data Availability

Not applicable.

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
