# Peer review of "Carotenoids: Role in Neurodegenerative Diseases Remediation"

_brainsci, 2023, doi:10.3390/brainsci13030457_

Round 1

Reviewer 1 Report

Carotenoids: Role in Neurodegenerative Diseases Remediation

The review is very interesting and complete. The authors explain all the defense and protection mechanisms against reactive oxygen species and redox stress that are involved in neurodegenerative diseases and other non-cerebral diseases, as well as the role of inflammation in neurodegeneration, aging and other age-related disorders and cancer.

They also reviewed the potential use of carotenoids in neurodegenerative diseases and their role as antiapoptotic agents in neuronal death.

The authors include a paragraph on clinical studies (page 12, line 365) and a table with the neurodegenerative disease pathway targeted by carotenoids. The title of this paragraph is confusing considering that the references cited in the text from 59 to 68 and the references in Table 2 are mostly from in vitro assays using bioactivity assays, cell culture experiments or on derived cell lines, which They are very important. and significant and show the activity or mechanisms of carotenoids as antioxidant, mitogenic, apoptotic agents and with great potential in neurodegeneration, but these cited studies do not refer to clinical studies in neurodegenerative diseases.

It would be advisable, in this paragraph named CLINICAL STUDIES, to separate the part of studies in vitro/in animal models from the studies in humans, where is described the activity of carotenoids in lung diseases, lung function or respiratory diseases as well as in lung cancer  (but not in neurodegenerative diseases).

The authors did a great job on this manuscript.

Reviewer 2 Report

This Manuscript highlights the role of carotenoids in neurodegenerative diseases. The authors provided a lot of information, however, not all of them were relevant. 

The Manuscript is not well structured. It is hard to understand it. There is an unnecessary repetition of the text throughout the paper and there is unnecessary jumping from one topic to the other and back to the first one. Relevant parts are missing. The authors should better divide chapters. Maybe it would be better if it started with types of neurodegenerative diseases, CTs and then accompanied with the role of CTs in the NDs.

Abstract: "Also referred to as degenerative brain disorders, NDs"-should be deleted since it has already been mentioned

Line 29: "during NDs"- should be deleted since it has already been mentioned

"OS takes place as a result of this increased production of reactive oxygen species."-should be deleted since it has already been mentioned

"A review of the literature on the relationship between OS,  neuroinflammation, and NDs has been assembled by us"- This sentence does not sound right.

"Protein aggregation is the root cause of neurodegenerative diseases (NDs), which are defined by the gradual death of neurons"- This is the first sentence in the Introduction, however, if protein aggregation is the root cause of NDs, why is it only briefly mentioned in the Abstract?

Line 47: Why "major", for a patient - any NDs is the major NDs?

Line 67: What are "dangers"?

Lines 96, 97,98, 99: There is a shifting from CTs to vitamins and then again to CTs, maybe it should be better to stick with one and then go to the other

Line 102: Shift from CTs to free radicals and OS and then again to CT

Line 119: Vitamins are mentioned in line 97, why are you mentioning it now again, why isn't it grouped?

-What is the connection between CT and vitamin E?

-Vitamin C is suddenly occurring, you never mentioned it before

Line 173: What is 3-NT?

Line 176: You never defined what is ROS and you introduced an abbreviation.

Line 186 and 188: Ursolic acid, TH-they just emerged, why are they mentioned, what are they and what are their roles?

Line 189: Histochemical method instead of histochemical should be written.

Line 195: Vitamin A has an abbreviation, but none of the other mentioned vitamins does not have it. All vitamins should be mentioned in the same way.

Line 196 and 198: Two exactly the same abbreviations, RA, only first one should be kept.

All vitamins should be grouped in one place.

Line 207: What do you mean by some ways?

Line 207 to 210: What is this all about, it is difficult to understand what is the purpose of these sentences and their role in the Manuscript.?

Line 212: Carotenoids should be CTs.

Line 217: You have already mentioned over 750 CTs, however, now it is written that there are more than 600. The authors should decide if is it more than 600 or more than 750.

Line 236: endogenous and exogenous, not foreign.

Line 248: These diseases are not characteristic only of the older population, they can occur, unfortunately, within the younger population too. 

Lines 256 and 261: Same types of inflammation are mentioned two times, acute and chronic inflammation, unnecessary repeating of the text.

Line 258: By suggesting that something lasts briefly, it already means it doesn't last long.

Line 271: You defined Nrf2, but not the NF-kB.

Line 278, 288, 289: What are Keep, sMaf, ARE, they are not defined in the Manuscript?

Table 1: IgA? Not defined.

Line 299: What are "these" disorders?

Lines 307, 308: tBid, Bcl-2 there is no explanation of what it is.

Lines 316-319: What is the meaning of this sentence?

Line 322: When? How, why?

Lines 339, 340: Why mentioning NDs, when this segment is only about AD? 

Lines 347,348,349: You have already stated that several times.

Lines 353, 366: You have already stated that several times.

Chapter 6. Clinical studies-Why is this whole chapter relevant to NDs? What is the purpose of it? What is the connection between the lungs, liver and brain?

Round 2

Reviewer 2 Report

The author's have answered to all of my questions and have addresses issues that were raised. The Manuscript now fits better for publication.